# Are Mediterranean Island Mountains Hotspots of Taxonomic and Phylogenetic Biodiversity? The Case of the Endemic Flora of the Balearic Islands

**DOI:** 10.3390/plants12142640

**Published:** 2023-07-13

**Authors:** Moisès Guardiola, Llorenç Sáez

**Affiliations:** 1Unit of Botany, Department of Animal and Plant Biology and Ecology, Universitat Autònoma de Barcelona, ES-08193 Bellaterra, Spain; 2Systematics and Evolution of Vascular Plants (UAB)—Associated Unit to CSIC by IBB, Unit of Botany, Department of Animal and Plant Biology and Ecology, Universitat Autònoma de Barcelona, ES-08193 Bellaterra, Spain; gymnesicum@yahoo.es; 3Societat d’Història Natural de les Balears (SHNB), Margarida Xirgu 16, ES-07003 Palma de Mallorca, Spain

**Keywords:** conservation, endemism, biodiversity hotspots, nano-hotspots, phylogeny, mountain flora, EDGE, phylogenetic diversity

## Abstract

The Mediterranean islands are exceptionally rich in endemism, most of which is narrowly distributed. Conservation measures, such as protected areas, have been prioritised, mainly on the basis of species richness and endemism, but phylogenetic information should also be taken into account. In this study, we calculated several taxonomic and phylogenetic metrics at a high resolution for the endemic flora of the Balearic Islands (154 taxa), in order to identify (i) the spatial patterns and environmental factors that explain this endemism, (ii) hotspots of species and phylogenetic endemism, and (iii) gaps in the protected areas. The taxonomic and phylogenetic metrics showed different distribution patterns, but the mountainous areas of Mallorca, and some coastal areas of the Balearic Islands, have the highest values. These values were positively related to elevation, precipitation, temperature, and slope, and negatively related to the distance from the coast, aspect, and the temperature of the wettest quarter. We identified top grid hotspots where all the metrics had the highest values, and we also identified nano-hotspots within these hotspots, in some of the highest peaks of Mallorca, where most of these metrics’ maximum values coincided. This approach allowed us to identify some gaps in the conservation priority areas, and to highlight the need to review their boundaries and definition.

## 1. Introduction

The Mediterranean Basin is one of the world’s biodiversity hotspots, due to its high plant-species richness and endemism [1,2,3]. This extraordinary biodiversity is the result of a complex biogeographical history, particularly influenced by its current and past geology, climate, and human influence, which has resulted in a high degree of environmental heterogeneity, and a patchy landscape, with different niches in a narrow extent [4,5]. All of these factors have contributed to the presence of high rates of narrow endemism in many regions of the Mediterranean Basin. Narrow-endemic taxa have a very restricted area of distribution, with at least 90% of their occurrences at, or below, the provincial level [6]. They are more prone to be threatened, because of their isolation and reduced distribution, habitat specificity, small populations, low competitive ability, and low genetic diversity, and around two-thirds are threatened, according to the IUCN criteria [6,7]. Narrow endemics in the Mediterranean Basin are mostly found on islands and in mountains [8,9], as these have acted as glacial refugia [10,11], and have favoured plant genetic radiation, hybridisation, introgression, and changes in the reproductive system [4].

Therefore, endemism rates are not uniform across regions, and can be detected at different spatial scales. There are also hotspots of plant endemism within the Mediterranean Basin hotspot [8], or hotspots within hotspots, with narrow or nano-hotspots that have a very small area, but a high proportion of the regional endemics [12,13]. However, there is no consensus on how to quantitatively detect hotspots on each spatial scale, and there are subjective thresholds, generally ranging from 1% to 10% of the richest grid cells, or the grid cells with more than 5% to 40% of the total regional endemics [3,8,12,13,14,15,16,17,18,19,20].

Conservation efforts, such as protected areas, have traditionally been defined on the basis of the species richness and endemism, because these are relatively easy to measure, and there are several sources of past and current information. However, the species richness does not take into account the evolutionary history and phylogenetic diversity of species [21]. The phylogenetic diversity is the amount of evolutionary history or evolutionary relationships that exist between species in a community or an area, and can be measured as the total branch length of the phylogenetic tree that includes all species in this community or area [22,23,24]. As a result, two communities with equal species richness may differ greatly in their phylogenetic diversity. In this sense, several studies in the last decade have pointed out that almost as important as the species richness, is the species phylogenetic diversity [25,26,27], and that both measures of biodiversity should be considered and integrated into conservation strategies and hotspot detection [6,18,22,24,28,29,30]. Nonetheless, research publications have focused mainly on species richness or species diversity, and have rarely considered phylogenetic diversity [31], although the spatial patterns of taxonomic and phylogenetic diversity may differ [17,28,32,33]. In addition, the development of other phylogenetic metrics in recent years has increased the knowledge of the phylogenetic information that should be incorporated into conservation strategies [22,24,31,34,35,36,37,38]. For example, phylogenetic endemism accounts for the range size of each taxon [30,36], and the EDGE (Evolutionarily Distinct and Globally Endangered) method [39] combines the relative contribution of each species to phylogenetic diversity (namely evolutionary distinctiveness), and its extinction risk using the IUCN (International Union for Conservation of Nature) Red List Categories, and produces a list of conservation priorities, including the species that are both evolutionarily distinct and globally endangered.

Taxonomic and phylogenetic metrics and endemism patterns are scale-dependent, and the spatial grain size used in the analysis of diversity metrics is of critical importance in detecting hotspots of taxonomic and phylogenetic endemism [18,40,41], as it modifies the relationship between the environmental and phylogenetic diversity variables [40,42]. Furthermore, not detecting all species in a given area can lead to a significant underestimation of the values of the species-richness and phylogenetic-diversity metrics [43].

The current implementation of protected areas, such as the European Natura 2000 network, has been based mainly on the presence of endemic or threatened species richness (Habitats Directive, Council Directive 92/43/EEC), but there is a lack of knowledge about important areas of phylogenetic diversity in protected and non-protected areas [6,31].

In this study, we investigated the patterns in endemic species richness and phylogenetic diversity in the Balearic Islands, an archipelago located in the western Mediterranean Basin that includes four main islands (Mallorca, Menorca, Eivissa, and Formentera), and approximately one hundred and fifty islets. The Balearic Islands, with approximately 6.9 to 10.4% of endemic species (including subspecies), comprise one of the regional hotspots of endemism in the Mediterranean Basin [44,45,46,47,48], and have the highest number (15 species) of extremely narrow endemics (taxa that occur in one, or few than five populations, and with fewer than five hundred reproductive and vegetative individuals [7]) in the Mediterranean Basin [6], and nine species included in the Top 50 Mediterranean Island Plants, curated by the Mediterranean Plant Specialist Group of the UICN [49].

We integrated a comprehensive distribution dataset at a high spatial resolution (1 km^2^) for all the endemic vascular plants of the Balearic Islands (154 narrow endemic species) with a phylogeny, and calculated several diversity measures. Our objectives were (i) to identify the spatial patterns of species and the phylogenetic diversity of the Balearic endemic vascular flora, (ii) to determine which environmental factors explain these patterns in species and phylogenetic endemism, and (iii) to identify hotspots of species and phylogenetic endemism, and gaps in the priority conservation areas.

## 2. Materials and Methods

### 2.1. Study Area

The Balearic archipelago, located in the western Mediterranean Basin, comprises four main inhabited islands: Mallorca (3640 km^2^), Menorca (701 km^2^), Eivissa (541 km^2^), and Formentera (82 km^2^) (Appendix A). Additionally, there are approximately 150 uninhabited islets, but the most relevant is the Cabrera sub-archipelago (13 km^2^). Mallorca is a mainly calcareous island, consisting of three geomorphological units: the Serra de Tramuntana (hereafter ST), the Serra de Llevant (hereafter SL), and the central lowlands. The ST is a mainly karstic mountain range, extending 90 km along the northwest of the island. There are several mountains over 1000 m, the highest being Puig Major (1445 m). The SL, in the southeast, has a less marked topography, rising to 400–560 m. Menorca has no high mountains (maximum height 358 m), and the northern part has non-calcareous rock formations and gentle hills that contrasts with the calcareous south of the island, which is crossed with deep ravines. Eivissa is, like Formentera, an entirely calcareous island, and has a varied topography, with mountainous areas located in the north and southwest that reach 475 m elevation. Formentera is the flattest of the main islands (maximum height 202 m). The climate of the Balearic Islands is Mediterranean, with a dry summer. Rainfall increases from west to east, and from south to north. However, the rainfall is also strongly related to the topography. In northern Mallorca (ST), the maximum precipitation can exceed 1500 mm/year in the highest mountain areas, whereas the minimum, on the coast, is lower than 300 mm/year [50].

The phytogeography and vegetation of the Balearic archipelago reflects the complex paleogeographic and climatic events that took place during the Tertiary and Quaternary periods, in the western Mediterranean. Two distinct biogeographical units are distinguishable [51]: the eastern or Gymnesian islands (Mallorca, Menorca, and Cabrera), with strong floristic Tyrrhenian affinities, and the western or Pityusic islands (Eivissa and Formentera), which possess a clear Iberian affinity. However, Fenu et al., (2020) [48], analysing the threatened island vascular plants in the Mediterranean Basin, included all the Balearic Islands in the Tyrrhenian group; and Buira et al., (2021) [44], analysing the spatial patterns in Iberian and Balearic endemic vascular flora, found that both island groups form a single differentiable endemic unit.

### 2.2. Species Occurrence Data

Firstly, we reviewed and compiled all endemic vascular plant species and subspecies (hybrids were omitted) of the Balearic Islands (Appendix A). For the endemic flora of the Balearic archipelago, we essentially followed the compilation of [52], updated according to Castroviejo (2021) [53] and recent scientific papers. Taxa of uncertain systematic value have been excluded (Appendix A). Taxa were considered endemic whenever their native distribution was limited to the Balearic archipelago. However, in a few cases, stenoendemic (or subendemic) taxa occurring in adjacent geographic areas were included, but only if they had most (more than 50%) of their distribution area in the Balearic Islands (*Asplenium majoricum* Litard., *Carduncellus dianius* Webb, *Cyclamen balearicum* Willk., *Diplotaxis ibicensis* (Pau) Gómez-Campo, *Medicago citrina* (Font Quer) Greuter, *Micromeria filiformis* (Aiton) Benth., *Romulea columnae* subsp. *assumptionis* (Font Quer) O. Bolòs, Vigo, Masalles & Ninot, *Silene cambessedesii* Boiss. & Reut. and *S. hifacensis* Willk.). The narrow-endemic *Lysimachia minoricensis* J.J.Rodr. was excluded, because it is extinct in the wild [52].

Secondly, all the distributional information, at 1 km resolution, was retrieved from trustable databases available from electronic repositories (Bioatlas of the Ministry of Environment and Territory of the Balearic Islands: https://bioatles.caib.es (accessed on 19 May 2022), herbarium data, and published research papers, including an unpublished database composed of over 30 years of field data compiled by LS. A total of 17,832 endemic plant occurrence records were compiled (Appendix A) and displayed in a 1 × 1 km^2^ grid resolution for all the Balearic Islands, including a total of 16,119 squares.

### 2.3. Environmental Data

We used topographic and climatic variables to explain the endemic species richness and phylogenetic diversity. The elevation was obtained from the digital terrain model MDT25 from the Spanish National Geographic Institute (www.ign.es) (accessed on 19 May 2022), obtained by interpolation from the LIDAR flights of the first National Aerial Orthophotography Plan (PNOA), with a resolution of 25 cm/pixel. The slope, aspect, and distance from the coast were derived from the digital terrain model MDT25, using QGIS 3.24 [54]. The climatic variables were downloaded from the CHELSA database [55] at high resolution (30 arc sec, approximately 1 × 1 km^2^); this database includes 19 bioclimatic variables. The mean value of each variable was calculated using a national reference grid of 1 × 1 km^2^ from the Spanish National Geographic Institute (projection ETRS89 UTM Zone 30N).

We performed a correlation analysis of all these environmental variables, and selected those that were not highly correlated (Spearman’s rank correlation < 0.7), to reduce multicollinearity and model overfitting. We also removed the bioclimatic variables with very low variability in the Balearic Islands (i.e., Bio3 with only three values: 27, 28, and 29; Bio2 with five values: 52 to 56; etc.). Finally, seven environmental variables were selected (Appendix A): the annual mean temperature (Bio1), mean temperature of the wettest quarter (Bio8), annual precipitation (Bio12), elevation, aspect, slope, and distance from the coast.

### 2.4. Phylogenetic Data

To calculate the phylogenetic diversity metrics of Balearic endemic plants, we constructed a phylogenetic tree, by linking the list of plants with a dated megaphylogeny of seed plants, using the package V.PhyloMaker [56]. This package uses the dated mega-tree for seed plants by [57,58] for pteridophytes. A few species were not included in the phylogeny, so we changed the species names to other sister species, or changed them to other near-synonym taxa included in the phylogeny (i.e., *Coristospermum huteri* (Porta) L. Sáez & Rosselló to *Ligusticum lucidum* Mill., *Helosciadium bermejoi* (L. Llorens) Popper & M.F. Watson to *Helosciadium nodiflorum* (L.) W.D.J.Koch, *Carduncellus dianius* to *Carthamus dianius* (Webb) Sch.Bip. or *Mauranthemum ebusitanum* (Vogt) N. Torres & Rosselló to *Leucanthemum monspeliense* (L.) H.J.Coste) and, once the phylogeny was built, we changed them back to having the correct name. The merged phylogeny included all 154 endemic species, and 130 internal nodes (Figure 1; Figure 1). It should be noted that there were some, or many, polytomies in the phylogeny for some rich genera, such as *Limonium* Mill., *Euphorbia* L., *Teucrium* L., or *Genista* L. This is a common caveat in phylogenetic analysis; however, ref. [59] pointed out that trees with polytomies are highly correlated to those constructed from a phylogeny resolved at the species level, and that the relationships between geographical and ecological gradients, with the derived phylogenetic metrics of both phylogenetic trees, are comparable.

### 2.5. Diversity Measures

We calculated the endemic species richness (RE) per 1 km^2^, using the data compiled by [52], the Bioatlas https://bioatles.caib.es (accessed on 19 May 2022) of the Ministry of Environment and Territory of the Balearic Islands, and personal observations and the latest information about endemic species updated up to 2022 by one of the authors (LS). We also calculated the weighted endemism (WE) and the corrected weighted endemism (CWE) at each grid. The WE is the sum of the endemic species at each grid weighting by the inverse of the species distribution range, and the CWE is the WE divided by the species endemic richness of each grid [41,60]. Using the compiled endemic species richness data and the constructed phylogenetic tree, we calculated the phylogenetic diversity (PD) and the phylogenetic endemism (PE) at each 1 km^2^ grid. The PD is the amount of evolutionary history or the evolutionary relationships between the species in a community or an area, and can be measured as the total branch length of the phylogenetic tree including all the species of this community or area [22,23,24], while the PE takes into account the range size of each taxon [30,36]. As the PD is not independent from the species richness (RE), we computed the standardised effect size for the PD (sesPD), dividing the difference between the observed and expected PD values by the standard deviation of the null distribution, with 1000 replications shuffling the taxon labels in the phylogeny, with the *PD_ses* function implemented in the ‘phyloregion’ package [61]. Any sesPD values (Z-scores) below and above −1.96 and +1.96 were, respectively, considered significant phylogenetic clustering and overdispersion [62]. Clustering occurred when closely related species occurred in the same grid, whereas overdispersion occurred when distant species appeared in the same grid.

We also calculated the EDGE value for all species. This metric combines the evolutionary distinctiveness (ED) of each species with the global endangerment (GE) of the species, according to the IUCN categories [39]. The ED was calculated by dividing the total phylogenetic diversity of a clade amongst its members, using the *evol.distinct* function in the *picante* package [63], following Isaac et al., (2007) [39]. The IUCN categories were extracted from [52], with some updates (Appendix A), and the GE scores were coded as [39]: critically endangered = 4, endangered = 3, vulnerable = 2, near threatened = 1; least concern = 0; DD species were excluded (*Limonium bolosii* Gil & L. Llorens, *L. escarrei* L. Llorens & Tébar and *Magydaris pastinacea* subsp. *femeniesii* O. Bolòs & Vigo). The EDGE score for each species was calculated as: EDGE = ln(1 + ED) + GExln(2) [39], using the *EDGE* function in the *phyloregion* package [61]. Finally, the EDGE species values per grid were summed, to obtain an EDGE value per grid cell [16,19].

### 2.6. Data Analysis

Prior to any analysis, all the response variables (the RE, WE, CWE, PD, and PE) except sesPD were log10 transformed, to reduce the skewness of their distribution. To detect the relationships between these variables, we calculated the Spearman’s correlation coefficient for each pair. The environmental and response variables were centred and scaled (standardised to zero mean and unit variance), to make their parameter estimates comparable. We fitted generalised linear models (GLMs) with Gaussian distribution with each response variable (the RE, WE, CWE, PD, PE, and sesPD), and all environmental variables as predictor variables (Bio1, Bio8, Bio12, the elevation, aspect, slope, and distance from the coast), in order to understand how endemic species richness and phylogenetic diversity were distributed in the Balearic Islands. With all the models, we performed a model selection approach to select the best models, using the Akaike information criterion (with ∆AIC < 2; [64]) for the model-averaging procedure. The model averaging was performed using MuMIn package 1.46.0. [65]. All statistical analyses were performed using R version 4.2.2 [66].

### 2.7. Conservation Priority Areas

To identify the conservation priority areas or hotspots in the Balearic Islands, we summed all the species richness and phylogenetic diversity metrics as follows. Firstly, we selected the grid cells with the highest 1% values for RE, WE, CWE, PD, PE, sesPD, and EDGE. Secondly, we calculated how many of these top 1% metrics appeared at each 1 × 1 km^2^ grid square (ranging from 0 to 7). We repeated this procedure for the top 2.5%, 5%, and 10% in all the species richness and phylogenetic diversity metrics. Squares with a value of 7 (for the top 1%, 2.5%, 5%, or 10% values) indicated a hotspot for all aspects of species richness and phylogenetic diversity, while squares with a value of 1 indicated a hotspot for only one metric. Finally, we overlapped the Natura 2000 network and the local protection network sites (available at https://www.caib.es/sites/espaisnaturalsprotegits (accessed on 19 May 2022)) with the obtained conservation priority areas, in order to identify gaps in the endemic vascular plant hotspot protection.

## 3. Results

The endemism in the Balearic Islands includes 154 (Appendix A) species; 145 are strictly endemics and 10 are subendemics (with the majority of the world population in the Balearic Islands). These 154 species include 39 subspecies, and belong to 91 genera and 37 families. With 24 species, the most diverse genus is *Limonium*, followed by *Euphorbia* and *Teucrium* with 5 species, *Genista* and *Silene* with 4 species, *Allium* L., *Galium* L., and *Thymus* L. with 3 species, *Anthyllis* L., *Arenaria* Ruppius ex L., *Brimeura* Salisb., *Carduncellus* Adans., *Cephalaria* Schrad., *Chaenorhinum* (DC.) Rchb., *Helichrysum* Mill., *Hieracium* L., *Hippocrepis* L., *Hypericum* Tourn. ex L., *Micromeria* Benth., *Ononis* L., *Polycarpon* Loefl., *Ranunculus* L., *Rhamnus* L., *Rubia* L., *Santolina* L., and *Viola* L. with 2 species, and the other 67 genera with only 1 species.

### 3.1. Spatial Patterns of Species Richness

The endemic species richness (RE) was higher on the island of Mallorca (Appendix A), with 109 endemic species. This was highest in the mountainous areas of the ST, located in the northwest of the island, and in the SL, located in the northeast. Both zones had some squares with more than 26 endemic species, and a maximum of 54 species per square in the ST, and 39 in the SL. Menorca was the second-richest island, with 58 endemic species, and with two 1 × 1 km^2^ squares with a maximum of 21 endemic species located in the S’Albufera des Grau, in the northeast of the island. Eivissa was the third-richest island, with 39 endemic species, and with a maximum of 17 species per 1 × 1 km^2^ square in the north of the island. The islands of Cabrera and Sa Dragonera were the joint fourth-richest islands, with 28 endemic taxa each, and with a maximum of 21 and 17 endemic species per 1 × 1 km^2^ square, respectively. Finally, Formentera was the poorest island, with 19 species, and with a maximum of 8 species per 1 × 1 km^2^ square in the north of the island. The distribution of endemic species richness per island follows the endemic-species-richness–area relationship (adj. R^2^ = 0.92, *p*-value = 0.0013), but with Menorca below the expected value, and Mallorca and Sa Dragonera with slightly higher values. All environmental variables contributed to explaining the RE, except for the aspect (Figure 2; Table 1). The elevation was the strongest predictor of RE (model-averaged coefficient 0.233; Figure 2; Table 1), followed by the distance from the coast (−0.196), Bio12 (0.158), Bio1 (0.153), slope (0.148), and Bio8 (−0.092). The weighted endemism (WE) ranged from 0 to 3.81, and was more evenly distributed among the islands (Appendix A). The highest values (WE > 1) were located in Mallorca (a total of 15 1 × 1 km^2^ squares: 7 in the ST in the north of the island, and isolated points on the west and east coasts), two squares on the north and west coasts of Eivissa, and one square on the east coast of Menorca. All the environmental variables contributed to explaining the WE (Figure 2; Table 1): the elevation was the strongest predictor of the WE (−0.197), followed by Bio1 (0.153), the distance from the coast (−0.097), Bio12 (0.064), Bio8 (−0.053), the slope (0.019), and the aspect (−0.013). The corrected weighted endemism (CWE) ranged from 0 to 0.52, and was more evenly distributed among islands than the RE and WE (Appendix A). All the environmental variables except the aspect and Bio12 contributed to explaining the CWE (Figure 2; Table 1): the distance from the coast was the strongest predictor of the CWE (−0.082), followed by Bio1 (0.073), the elevation (0.069), Bio8 (−0.034), the slope (0.029), Bio12 (−0.007), and the aspect (−0.006).

### 3.2. Spatial Patterns of Phylogenetic Diversity

The phylogenetic diversity (PD) was highly correlated with the RE (r = 0.72), and showed a similar geographical distribution to the RE (Appendix A), with maximum values in the mountainous areas of northern Mallorca (the ST and SL), and medium values surrounding the islands of Menorca, Eivissa, and Formentera. All the environmental variables contributed to explaining the PD, except Bio1 (Figure 2; Table 1). The distance from the coast was the strongest predictor of PD (−0.299; Figure 2) (Table 1), followed by Bio12 (0.237), the slope (0.168), the elevation (0.119), the aspect (−0.109), and Bio8 (−0.071). The highest and lowest values of the standardised effect size for the PD (sesPD) were specially concentrated in the ST, and in a few points on the coasts of Mallorca, Menorca, and Formentera (Appendix A). However, a few cells significantly deviate from the expected PD under the null model, and show significant phylogenetic clustering, with significant low sesPD or overdispersion with significant high sesPD [62]. The zones with phylogenetic overdispersion were only located in a few squares in the mountainous area of the ST in Mallorca, while the zones with phylogenetic clustering were located in the northwest and in scattered squares along the coast of Mallorca, in some inner squares in Menorca, and in one square in northern Formentera (Appendix A). All the environmental variables except the slope and Bio8 contributed to explaining the sesPD (Figure 2; Table 1): Bio1 was the strongest predictor of the sesPD (0.765), followed by the elevation (0.752), Bio12 (0.258), the distance from the coast (−0.192), and the aspect (−0.06). The values of phylogenetic endemism (PE) were also higher in the ST in Mallorca, and medium values appeared in the SL in Mallorca, and in some points on the Menorca, Eivissa, and Formentera coasts (Appendix A). All the environmental variables contributed to explaining the PE (Figure 2; Table 1): the elevation was the strongest predictor of PE (0.253), followed by Bio1 (0.191), the distance from the coast (−0.155), Bio12 (0.136), Bio 8 (−0.062), the aspect (−0.053), and the slope (0.037).

At the species level, the EDGE values ranged from a minimum value of 2.107 for *Limonium biflorum¸ L. minoricense*, and *L. minutum*, to 5.494 for *Asplenium majoricum* (Appendix A). Threatened species according to IUCN categories [67] did not tend to have higher EDGE values (one way ANOVA, *p* = 0.8) (Figure 3). The EDGE values at the square level, summing all the EDGE values for each species present at each grid square, ranged from 2.1 to 200.6, with the maximum values at the Serra de Tramuntana, cap de Formentor, and Massís d’Artà, all on the island of Mallorca (Appendix A).

The species richness and phylogenetic diversity variables were low-correlated, except for the RE and PD (r = 0.72), the PE and PD (r = 0.69), and the WE and CWE (r = 0.64) (Appendix A).

### 3.3. Conservation Priority Areas

The mountainous areas of Mallorca, especially the Serra de Tramuntana, and to a lesser extent the Serra de Llevant, were the zones of the Balearic Islands that included the most hotspots for the RE, PD, sesPD, WE, CWE, PE (Appendix A), and EDGE values (Appendix A). The very same zones emerged when selecting the top 1%, 2.5%, 5%, and 10% hotspots in terms of these species richness and phylogenetic diversity variables (Figure 4a–d). In the ST, there were nine squares with five top 1% hotspots for these variables, eight squares with four top 1% hotspots, twenty-two squares with three top 1% hotspots, and several squares with one or two top 1% hotspots for these seven variables. In the SL, there were two squares with three top 1% hotspots of these variables, one square with two top 1% hotspots, and six squares with one top 1% hotspot. The other main islands in the Balearic Islands only had one or two top 1% hotspots. Eivissa had six squares with two top 1% hotspots, and eighteen squares with one top 1% hotspot, mainly located in the north and northwest of the island. Menorca had only one square with two top 1% hotspots, and thirty-nine squares with one top 1% hotspot, scattered all over the island. Formentera had only one square with one and two top 1% hotspots, respectively, in the north of the island. Finally, Cabrera had three squares with one top 1% hotspot, in the south of the island.

Considering the top 2.5%, 5%, and 10% hotspots for all the endemic species richness and phylogenetic diversity metrics (Figure 4b, Figure 4c, and Figure 4d respectively) reinforced the same hotspots detected accounting for the 1% hotspots, but other zones appeared as hotspots. In particular, in the case of the 10% (Figure 4d), the mountainous area in the north of Mallorca (the ST) was the zone that included the most hotspots for the species richness and phylogenetic diversity variables, followed by the SLL in eastern Mallorca, the north and southwest coast of Menorca, the south of Cabrera, and the north of Eivissa.

The top 1% of grids included almost one square containing 97.4% of all endemic Balearic species (149 species, except *Brimeura duvigneaudii* (L. Llorens) Rosselló, Mus & Mayol subsp. *duvigneaudii*, *Delphinium pentagynum* subsp. *formenteranum* N. Torres, L. Sáez, Rosselló & C. Blanché, *Limonium bianorii* (Sennen & Pau) Erben, *Orobanche iammonensis* Pujadas & Fraga, and *Polygonum romanum* subsp. *balearicum* Raffaelli & Villar), and with a mean for all species of 46.9% of squares (Table 2), where each species is present in the top 1% squares. This top 1% of squares include 100% of the CR species, 66% of the EN species, and 43% of the VU species. Within the top 2.5%, 5%, and 10% squares, all the endemic species were included, with a mean for the entire species distribution of 65.3% (100% of all CR species, 82% of EN species, and 69% of VU species), 79.2% (100% of all CR species, 92% of EN species, and 82% of VU species), and 88.1% (100% of all CR species, 94% of EN species, and 86% of VU species), respectively (Appendix A).

## 4. Discussion

We present an analysis of the species richness and phylogenetic diversity of the entire endemic flora of the Balearic Islands, on a high-resolution scale, based on the dataset of a 1 × 1 km^2^ grid, including environmental variables and comprehensive species occurrence data. We used a detailed scale that is rarely used in studies [12,42,68,69], even though coarse datasets or low-resolution datasets can affect the detection of hotspots of species richness and phylogenetic diversity metrics [40,43,70], or modify the relationship between environmental and species richness and phylogenetic variables [40,42]. Our results highlight the importance of the mountainous areas of northern Mallorca for different facets of endemic biodiversity, and the ways in which climatic and topographic drivers explain each metric related to species richness or phylogenetic diversity. The elevation, annual precipitation (Bio12), annual mean temperature (Bio1), and slope always had a positive effect on the RE, PD, sesPD, WE, CWE, and PE, while the distance from the coast, the aspect, and the mean temperature of the wettest quarter (Bio8) had negative effects. Our analysis, using these metrics of species richness and phylogenetic diversity, has allowed the identification of several hotspots and nano-hotspots of endemism in the Balearic Islands.

### 4.1. Spatial Patterns of Species Richness

The vascular flora of the Balearic Islands includes about 1551 species [47], and we compiled the presence of 154 endemic species (Appendix A), which correspond to 9.9% of endemism. This rate of endemism is similar to, or higher than that reported in previous studies (10% in [46], 6.9% in [45], and 7.5% in [44]), and higher than on other Mediterranean islands (6.9% in Sardinia, but with 168 endemic taxa [71]; 5.5% in Corsica, with 122 endemic taxa [72]; and 8.7% in Crete, with 174 endemic taxa [73]). The highest levels of endemism in the Mediterranean islands and other Mediterranean hotspot areas are located in the mountain ranges, although the total species richness decreases with elevation [5,12,13,44,68,72,74,75,76,77,78,79]. We also found that the centres of RE in the Balearic Islands were in the highest mountainous areas: the Serra de Tramuntana and Serra de Llevant, located in the north and east of the island of Mallorca, respectively (Appendix A). Mountain belts in the Mediterranean isles harbour the maximum rate of endemism; this decreases with the descent towards the Mediterranean belt, and the ascent towards the Subalpine belt [5,72]. The Serra de Tramuntana is the highest area in the Balearic Islands, and includes 52% of all the Balearic endemics; 25% of the Balearic endemics are restricted to this zone [47]. According to Steinbauer et al., (2013) [74], the increase in endemics richness with elevation is due to an elevation-driven ecological isolation caused by the distance from other similar habitats, refugia for mountain species during climatic fluctuations, and species-poor ecosystems, and is also because lowland sites are connected to the continent, increasing the propagule pressure from the continent. The latter is true, except for the case of rocky coastal habitats, as we found that the distance from the coast was negatively related to the RE (Table 1). Aside from mountain systems, some of the coastal areas in the Balearic Islands had cliffs and rupicolous habitats which were also rich in endemics: the ST and SL in Mallorca, northern Menorca, and Eivissa (Appendix A). In contrast, the islands’ inner territories were poor in endemic species. Cliffs are a harsh environment, with a relative stable climate over time, have less biotic competition due to open vegetation and low total species richness, and are far from human impacts, so endemic species can persist in such habitats [5,9]. The cliffs and rupicolous habitats of the mid-altitude Iberian Peninsula mountains act as a refugia, and harbour the highest endemicity richness, especially in the Balearic Islands [80]; moreover, most endemics have evolved in that habitat [81]. Climatic variables also explained the RE (Table 1): the annual precipitation (Bio12) and annual mean temperature (Bio1) did so positively, and the mean temperature of the wettest quarter (Bio8) did so negatively. Precipitation and temperature are key factors in endemism [82], and several authors found that on the Mediterranean Islands, and in other Mediterranean hotspot areas, precipitation increases the endemic richness [12,13,68,83]. Finally, we found that slope was the last variable that explained the RE in the Balearic Islands (Table 1). Buira et al., (2021) [44] found that elevation and slope were the best predictors to explain the endemic species richness flora in the Iberian Peninsula and the Balearic Islands. Steep slopes increase endemism, because they are a high-stress habitat, with low biotic competition, habitat isolation, and a low human impact [68].

The Balearic Islands are a hotspot of weighted endemism (WE), due to their isolation, mountain landscapes, and climatic variables [44,80], and the presence of narrow endemics [84]. Our closer look at the Balearic Islands shows that some mountain ranges have the highest WE and RE values, but there are also some isolated UTM squares on coastal dunes or cliffs (Appendix A). The corrected weighted endemism (CWE) has a different behaviour to the WE. The highest values were located at some parts on the coast of the main Islands (Appendix A), with UTM squares that had high percentage of endemics restricted to a square, but not necessarily with a high species richness. The CWE is known to be better in detecting biodiversity hotspots even if the RE is not very high, because species with a wide range can mask the WE values [13,60,85]. In both the mountainous and coastal areas of the Balearic Islands, we found the narrowest range-restricted species present in only one grid (*Agrostis barceloi* L. Sáez & Rosselló, *Brimeura duvigneaudii* subsp. *occultata* L. Sáez, Rita, Bibiloni, Roquet & López-Alvarado, *Euphorbia margalidiana* Kuhbier & Lewej., *Helianthemum scopulicolum* L. Sáez, Alomar & Rosselló, *Helosciadium bermejoi, Limonium barceloi* Gil & L. Llorens, *L. bolosii, L. carvalhoi* Rosselló & L. Sáez, *L. ejulabilis* Rosselló, Mus & Soler, *L. escarrei, L. pseudodyctiocladon* L. Llorens, *Santolina vedranensis* (O. Bolòs & Vigo) L. Sáez, M. Serrano, S. Ortiz & R. Carbajal and *Taraxacum majoricense* Galán & L. Sáez), or two or three UTM squares (*Euphorbia fontqueriana* Greuter, *Limonium boirae* L. Llorens & Tébar, *L. inexpectans* L. Sáez & Rosselló, *L. leonardi-llorensii* L. Sáez, Carvalho & Rosselló, *L. magallufianum* L. Llorens, *L. marisoliis* L. Llorens, *L. migjornense* L. Llorens, *L. wiedmannii* Erben, *Thymus herba-barona* subsp. *bivalens* Mayol, L. Sáez & Rosselló with two; *Arenaria bolosii* (Cañig.) L. Sáez & Rosselló, *Avellinia longiaristata* (Font Quer) Romero Zarco & L. Sáez, *Cephalaria squamiflora* subsp. *ebusitana* (O. Bolòs & Vigo) O. Bolòs, *Coristospermum huteri* and *Cotoneaster majoricensis* L. Sáez & Rosselló with three). However, the RE and WE values were higher in mountainous areas than in coastal areas, while the CWE was the opposite. It should be noted that *Limonium* is one of the most species-rich and highly diversified Mediterranean plant lineages [13,44,84,86,87], with 24 endemics in the Balearic Islands, mostly narrow endemics that predominantly grow in coastal areas [52], and have recently diversified [75], thus contributing to the higher CWE values in coastal zones.

### 4.2. Spatial Patterns of Phylogenetic Diversity

We found that the phylogenetic diversity (PD) was highly correlated, and showed a similar geographical pattern to that of the RE (r = 0.72; Appendix A). The correlation between the PD and RE is a common pattern [14,24,26,33,88,89,90] but, as other studies pointed out [19,26,91], the PE and sesPD were not highly correlated with the RE (r = 0.50 and 0.23, respectively; Appendix A), and the sesPD was not correlated with the PD (r = 0.38; Appendix A), but was more correlated with the PE (r = 0.69; Appendix A), because the PE accounts for the spatial distribution of the species, and the sesPD is standardised according to species diversity. Hence, areas of high species richness are not always areas with a high phylogenetic diversity or phylogenetic endemism [26].

The two environmental variables that better explained the PD were the distance from the coast (negative relationship) and Bio12 (positive relationship) (Figure 2; Table 1). The PD is the sum of the total phylogenetic branch length in a grid; hence, grids with a higher PD are especially located in the mountainous zones of Mallorca in the ST and SL that are near the coast and have a high annual precipitation (Bio12). In contrast, the elevation and Bio1 were the two environmental variables that better explained the PE and sesPD (Figure 2; Table 1). The greater the elevation and Bio12, the higher values of the PE and sesPD, probably because the highest peaks of the ST in Mallorca have several extremely narrow endemics of phylogenetically distant genera. The PD hotspots were located mainly in Mallorca, while the PE and sesPD highlight new hotspot zones in the Balearic Islands that were not selected with the traditional species richness metrics, especially on the islands of Menorca, Eivissa, and Formentera. Phylogenetic diversity metrics can help in conservation planning, because they quantify different facets of evolutionary diversity that are not captured by other measures [19,24,26,27]. The PE accounts for range-restricted species with phylogenetic distinctiveness, and shows the areas to which a substantial proportion of phylogenetic diversity is restricted, usually with high PD values and a small area distribution [92].

When combining the evolutionarily distinct and the globally endangered values for all endemic species, we found that the EDGE values ranked from 2.1 for some *Limonium* species, to above 5.48 for two ferns (Appendix A), with a mean value of 3.6 for all endemic species, and a mean of 4.0 considering the top 100 values. The EDGE values of the whole Greek endemic flora, with 1384 endemic species, were slightly wider than those of the Balearic Islands, between 1.66 and 8.77 [93], with a mean of 4.25 for all species, and a mean of 6.4 considering the top 100 values (values calculated by the authors, using data from [13]). The highest EDGE values of the endemic Balearic flora were for the ferns *Asplenium majoricum* and *Dryopteris pallida* subsp. *balearica* (Litard.) Fraser-Jenk. Neither species is threatened (they are NT and LC, respectively) but are more basal in the phylogenetic tree. In contrast, several *Limonium* species have the lowest EDGE values, although some are threatened (*L. fontqueri* (Pau) Erben and *L. wiedmannii* VU; *L. boirae*, *L. ejulabilis*, *L. inexpectans*, and *L. magallufianum* CR), because is a rich genus, and most species are clumped at the top of the phylogenetic tree. In the case of the endemic Greek flora [13], the highest EDGE values also included some species in the basal groups (such as ferns and gymnosperms), or species in the highest IUCN category (CR) and very narrowly-distributed, while the lowest EDGE values were from species within rich genera (such as *Centaurea*, with 78 species; *Limonium*, with 77 species; *Campanula*, with 60 species; *Hieracium*, with 60 species; or *Silene*, with 54 species) whether threatened or not (in the LC, NT, VU, or EN category). Species-rich clades or genera tend to have low ED values and, although some species could be threatened, their overall EDGE value is low [94]. Moreover, even though an increase in a Red List category represents a doubling of the extinction risk [39], we did not find that threatened species had higher EDGE values (one way ANOVA, *p* = 0.8; Figure 3). Neither [39], [95], or [94] found differences among the EDGE values and IUCN categories in mammals, gymnosperms, or fishes, respectively. Although species with low ED scores tend to have low levels of extinction risk, threatened status alone does not guarantee high EDGE values. In fact, endangered species sometimes are, on average, less evolutionary distinct than lower-concern ones [96]. Despite this, only 9.1% of Balearic endemics are considered CR, 11% EN, 14.3% VU, 59.7% LC and NT, and 1.9% DD, while 46.1% of the Greek endemics are considered CR, 34.8% EN, 15.9% VU, and 3.2% LC and NT [93]. Forest et al., (2018) compiled the information about the EDGE values for all of the world’s gymnosperms, amphibians, birds, and mammals. Looking at the top 100 species (according to the EDGE value), they found a mean between 5.1 and 5.5 for gymnosperms, birds, and mammals, and slightly higher (5.8) for amphibians, a higher mean value than we found in the Balearic Islands, or in Greece [93]. However, the top 100 species of these groups included high evolutionary distinct linages that were phylogenetically isolated, some of them considered living fossils or highly endangered. When summing the EDGE species values per grid, the maximum values were located in Mallorca, at the Serra de Tramuntana, Cap de Formentor and Massís d’Artà. The summed EDGE values identify high concentrations of the most phylogenetically distinctive and most endangered species [19].

### 4.3. Conservation Priority Areas

In the Mediterranean Basin, there is an urgent need to detect biodiversity hotspots because, in the last decades, especially on islands, important changes have been detected in species diversity [97], and in functional and phylogenetic patterns [98]. It is predicted that most range-restricted species will decline in the coming decades [99], and the current nature reserves, mainly established according to the total species richness and the endemic threatened species richness, will not be sufficient to protect important areas of phylogenetic diversity. Our approach, combining the top values of the WE, CWE, PD, PE, sesPD, and EDGE metrics, allows us to detect the top 1%, 2.5%, 5%, and 10% hotspot grids for species richness and phylogenetic diversity in the Balearic Islands. Interestingly, the top 1% of grids included almost one square of the 149 species (97.4% of all Balearic endemics), with a mean of 46.9% of squares for each species (Table 2). All UTM squares with CR species were included in the top 1%, 2.5%, 5%, and 10% squares; for EN species, 66% of their squares were included in the top 1%, 82% in the top 2.5%, 92% in the top 5%, and 94% in the top 10% of squares. Regarding the VU species, 43% of their squares were included in the top 1%, 69% in the top 2.5%, 82% in the top 5%, and 86% in the top 10% of squares. These results show that on only 2.5% to 5% of the Balearic Islands’ surface, it is possible to include all endemic species, and a high proportion of their current distribution ranges, especially for the most threatened species. These hotspots of species richness and phylogenetic diversity are located mainly in the mountainous areas of northern Mallorca, in the ST, but also in the SL in eastern Mallorca, the north and southwest coasts of Menorca and Cabrera, and northern Eivissa (Figure 4a–d). These zones overlap greatly with those reported by Mus (1995) [100]. In addition, we can delimitate nano-hotspots within these hotspots [12,13]. Nano-hotspots are defined as the richest 1 km^2^ grid cells of endemic species richness, identified at a massif level [12], but we propose to extend this concept not only to the richest grids for species richness, but also to the richest grids for phylogenetic diversity and, if data are available, the richest grids for functional diversity. In summary, nano-hotspots should include the richest grids of the three biodiversity dimensions or facets: taxonomic, phylogenetic, and functional diversity. The Serra de Tramuntana is the top hotspot in the Balearic Islands, with the highest RE, WE, CWE, PD, PE, sesPD, and EDGE values but, at a close scale, it includes nine squares with five top 1% hotspots of these seven variables. These nine nano-hotspots are located on some of the highest peaks: Puig de Maçanella (three squares), Puig Major (two squares), Puig de les Moles (two squares), Roca Blanca (one square), and Puig de Galatzó (one square). These nano-hotspots are zones with a very small area, but a high concentration of species richness and phylogenetic diversity; each contained over 32 endemic species (more than 21% of all of the Balearic endemics), and all 9 km^2^ included 70 endemic species (45.5% of all endemics). Cañadas et al., (2014) [12] found similar values for the nano-hotspots in the Gennargentu massif in Sardinia, with nine nano-hotspots of 1 × 1 km^2^ located at the highest peaks that accounted for 19.8% of the island’s endemics, but only used plant species richness as a selection variable. On a broader spatial scale, Huang et al., (2012) [14] found that the top 5% ER, PD, PE, and WE grids included about 80% of the endemic woody seed plant diversity in China.

Finally, by overlapping the Natura 2000 network and local network protection areas with the top 1% of grids, we found that most of the hotspots and nano-hotspots in northern Mallorca were included in those protection networks; however, several grid squares in the coastal or inland zones of Mallorca, Menorca, and Eivissa were not included in any of the protection networks (Figure 5). The non-protected hotspots that we detected should be considered as future extensions of the protected areas, at least on the local level, and could be designed as smaller protected areas; for example, micro-reserves [101].

### 4.4. Caveats and Limitations of the Study

Although our dataset included a current detailed compilation and review of all the endemic vascular plant species of the Balearic Islands, some wide-range endemics, such as *Teucrium capitatum* subsp. *Majoricum* (Rouy) T. Navarro & Rosua (with 3636 grids), *Cyclamen balearicum* (788), *Sonchus willkommii* (Burnat & Barbey) Rosselló & L. Sáez (664), *Limonium minutum* (L.) Chaz. (660), *Micromeria filiformis* (Aiton) Benth. (554), *Hypericum balearicum* L. (535) or *Rubia balearica* (Willk.) Porta (509), are likely to be present in a few undetected grid squares. The latter is unlikely, but could be in some inner grids in Mallorca or Menorca with no endemism reported. However, this would not significantly change our results. On the other hand, the rarest endemics have been intensively prospected for decades, and are less likely to be undersampled in respect of their real distribution.

Our study included the latest data on new species and locations in the Balearic Islands, such as *Aira minoricensis* Fraga, Romero-Zarco & L. Sáez [102], *Silene migjornensis* L. Sáez, Guasp, P.P. Ferrer, López-Alvarado & Rosselló [103], *Taraxacum majoricense* [104], and *Thymus richardii* Pers. subsp. *richardii* [105]; however, the description of new species is expected in the coming years, and could slightly modify our results in the future.

On the other hand, we constructed a phylogenetic tree, by linking our plant list with a dated megaphylogeny of seed plants, available in Jin & Qian (2019) [56]. However, this procedure generally produces several polytomies, especially for some rich genera. Even though Qian & Jin (2021) [59] found that trees with polytomies were highly comparable to those constructed from a phylogeny resolved at the species level, the resolution of these polytomies should improve our phylogenetic metrics, and could better delimit hotspots and nano-hotspots.

Another interesting point is that the WE, CWE, and PE metrics are calculated considering the distribution range of each species [41,60]. However, the range size and the abundance of the species are not always related, especially in island plant endemics [80]. We used a very detailed scale, squares of 1 × 1 km^2^, but several range-restricted species that appeared in less than five grids are not threatened, due to their abundance in suitable habitats (e.g., *Aira minoricensis* is LC and appeared in five grids, *Avellinia longiaristata* is NT and appeared in three grids, *Bellium artrutxensis* P. Fraga & Rosselló is NT and appeared in four grids, *Limonium saxicolum* Erben is LC and appeared in four grids, and *Orobanche iammonensis* is NT and appeared in five grids). However, the opposite is also true, and there are some EN species with more than 10 grids (*Rubia caespitosa* (Font Quer & Marcos) Rosselló had 14, *Malva minoricensis* J.J. Rodr. had 13, *Teucrium cossonii* subsp. *punicum* Mayol, Mus Rosselló & Torres had 12, *Medicago citrina* had 11, *Vicia bifoliolata* J.J. Rodr. had 11, and *Genista dorycnifolia* subsp. *grosii* (Font Quer) Font Quer & Rothm. and *Ononis crispa* subsp. *zschakei* (F. Herm.) L. Sáez & Rosselló had 10 grids). The differences between the range size and abundance could slightly modify the WE, CWE, and PE metrics, and thus the selection of hotspots.

Finally, we did not include anthropogenic factors as an explanatory variable because, although they are more or less important in some Mediterranean islands [68], the available data on the spatial scale of our study only included some modern anthropogenic impacts. However, the Balearic Islands have some particularities that make it difficult to measure, quantify, and delineate the spatial extent of these anthropogenic impacts. For example, human settlement started within the period of 2350–2150 BC, causing the extinction of several species, the most famous the endemic cave goat *Myotragus balearicus* Bate, 1909 [106]. The islands had been used by pirates and corsairs, and colonised several times by other cultures, and as an extreme event, the Cabrera Island was used in its entirety as an open prison in 1809, after the surrender of General Dupont in Baylen, and about 9000 French soldiers were sent to the island of Cabrera, completely transforming its landscape [107]. Although the highest-elevation areas of the Balearic Islands have a very high concentration of endemic species, their conservation status is very uneven, and depends on several intrinsic and extrinsic factors. Initially, it could be assessed that the anthropic disturbance in the highest mountains would be lower compared to lower-altitude areas, but this cannot be generalized and, in some cases, there is even evidence to the contrary. The summit of Puig Major (the highest peak in the Balearic Islands) was blown up in 1958 to enable the installation of military radar facilities, and an access road was built. Although the Spanish Ministry of Defence has reduced the size of the radar installations, and is involved in the conservation of the natural heritage of the mountain [108], the road embankments and the blowing-up of the summit buried large areas, causing severe impacts on the populations of endemic and rare species [7,47,109]. On the other hand, in recent years, the presence of hikers and climbers in the highest areas of Mallorca has increased, which has caused the direct (voluntary or involuntary) destruction of specimens of threatened endemic species, as well as a loss of quality in the habitat. On the other hand, these higher-altitude areas have strong populations of introduced feral goats, which have a severe impact on the flora and vegetation of the mountainous areas of Mallorca [110,111].

## 5. Conclusions

The taxonomic and phylogenetic diversity metrics revealed different distribution patterns in the Balearic Islands: the ER and EDGE were higher in the mountainous areas of Mallorca (the ST and SL); the WE and PD were also higher in the mountain areas of Mallorca, as well as at points along the Mallorca, Menorca, Eivissa, and Formentera coasts; and CWE principally identified zones that harbour range-restricted species, mainly distributed on the islands’ coasts, but also in the mountains. The higher the RE, PD, WE, CWE, and PE values, the higher the elevation, annual precipitation, annual mean temperature, and slope, but the shorter the distance from the coast, and the lower the aspect and the mean temperature of the wettest quarter. In addition, by selecting the top 1%, 2.5%, 5%, and 10% squares according to these taxonomic and phylogenetic diversity metrics, we identified hotspots where most of these metrics coincide in a square. The mountainous areas of Mallorca (the ST and SL) were included as the top hotspots, along with several more isolated zones on the other islands. Within only the top 1% and 2.5% of grids, almost all endemic Balearic species were included (97.4% and 100%, respectively), and a high proportion of its current distribution (46.9% and 65.3%, respectively). We also identified nano-hotspots within these hotspots, as the richest grids of taxonomic and phylogenetic diversity metrics, where all or most variables coincide. These correspond to some of the highest peaks of Mallorca. Finally, some of the hotspots and nano-hotspots detected are not covered by any form of protection; as a result, we call for the protection of these exceptional areas rich in endemic taxonomic and phylogenetic diversity.

## Figures and Tables

**Figure 1 plants-12-02640-f001:**
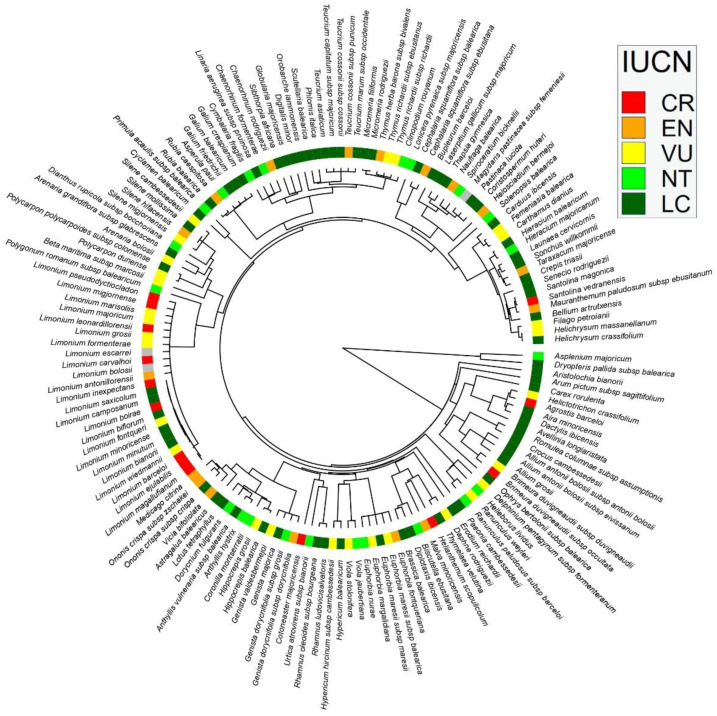
Phylogeny of Balearic endemic plant species, with their IUCN extinction risk categories: critically endangered (CR) = red, endangered (EN) = orange, vulnerable (VU) = yellow, near threatened (NT) = light green, least concern (LC) = green, data deficient (DD) = grey.

**Figure 2 plants-12-02640-f002:**
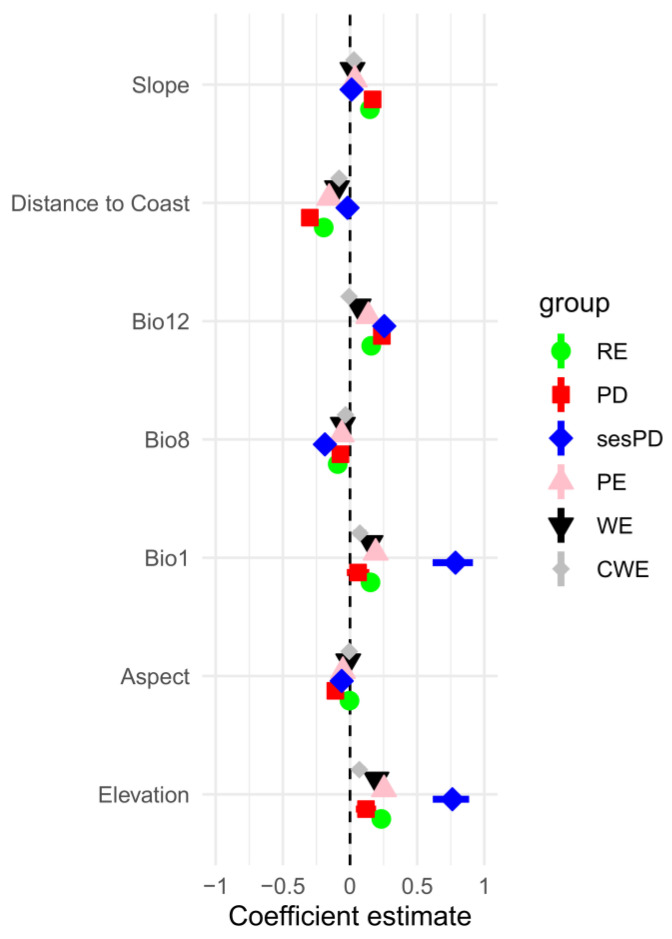
The model-averaged coefficient estimates of each environmental variable, with confidence-interval bars (standard error of the coefficient estimate) for the six diversity indices. As all the explanatory variables were standardised, each parameter estimate is comparable with the others. The points and confidence interval bars that overlap with the dashed line had a non-significant effect (see also Table 1). RE = endemic species richness, WE, CWE, PD, PE, sesPD, Bio1, Bio8, Bio12, elevation, aspect, slope, and distance from the coast.

**Figure 3 plants-12-02640-f003:**
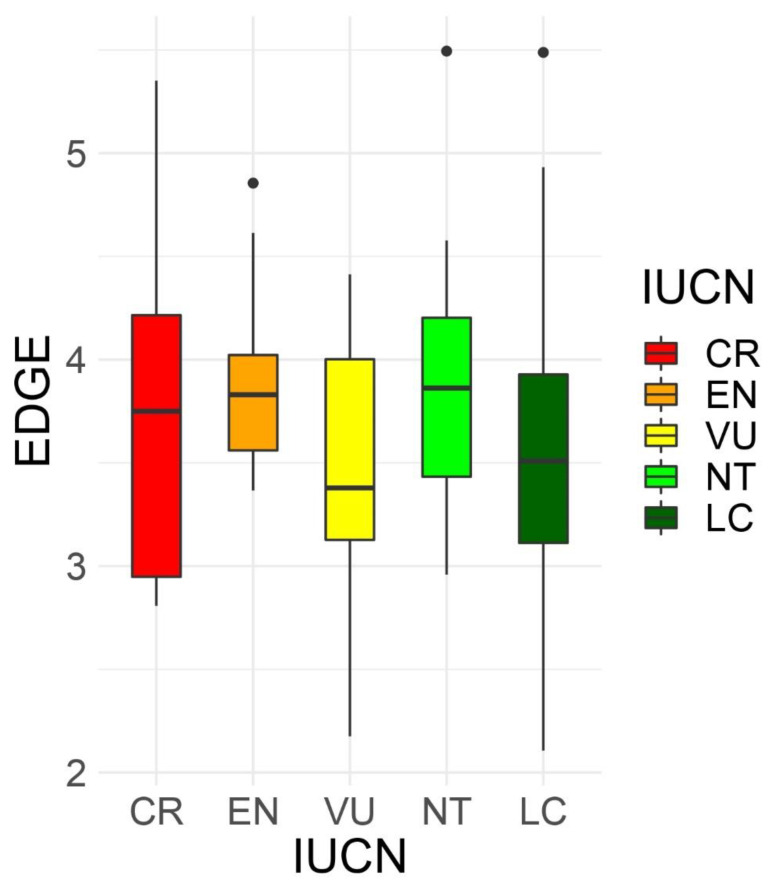
The boxplot of the distribution of the EDGE values of the endemic Balearic plant species, depending on their IUCN Red List category.

**Figure 4 plants-12-02640-f004:**
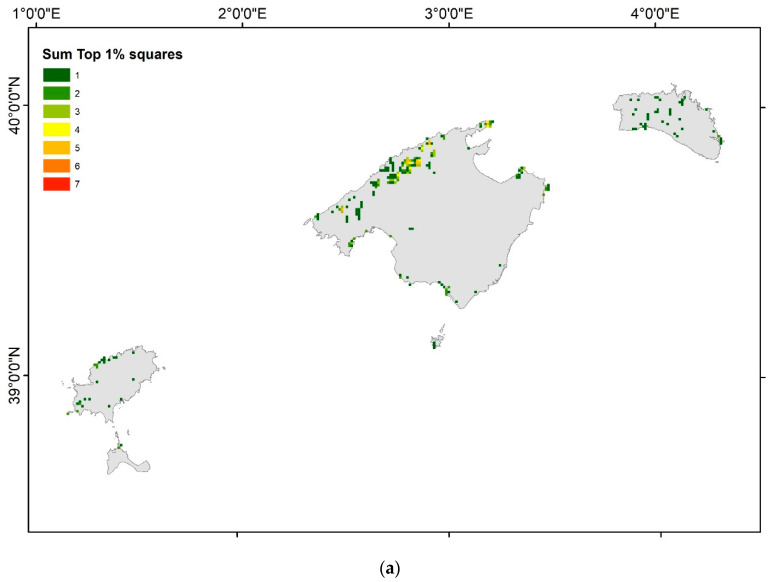
The sum of the UTM 1 × 1 km^2^ squares with the top (**a**) 1%, (**b**) 2.5%, (**c**) 5%, and (**d**) 10% of the values for RE, WE, CWE, PD, PE, sesPD, and EDGE.

**Figure 5 plants-12-02640-f005:**
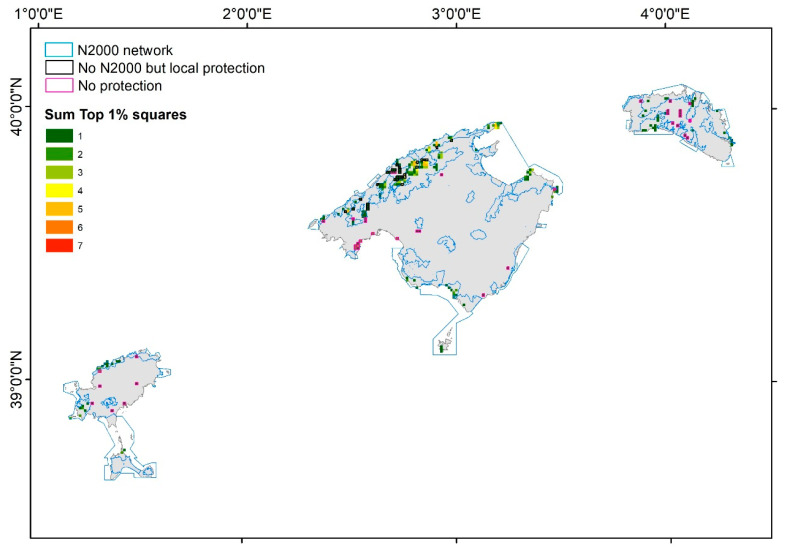
The top 1% of squares of the RE, WE, CWE, PD, PE, sesPD, and EDGE values. Blue lines denote areas included in the Natura 2000 network, grey squares denote squares included only in a local protection network, and pink squares denote squares included in any network.

**Table 1 plants-12-02640-t001:** The model-averaged coefficient estimates of each environmental variable for the six species richness and phylogenetic diversity metrics.

	Log(RE)		Log(WE)		Log(CWE)		Log(PD)		sesPD		Log(PE)	
	Estimate	Pr(>|z|)	Estimate	Pr(>|z|)	Estimate	Pr(>|z|)	Estimate	Pr(>|z|)	Estimate	Pr(>|z|)	Estimate	Pr(>|z|)
Intercept	−0.2381	0.0000	−0.1070	0.0000	−0.0992	0.0000	−0.4475	0.0000	−0.0508	0.001	−0.2049	0.0000
Elevation	0.2330	0.0000	0.1968	0.0000	0.0694	0.0038	0.1189	0.0020	0.7516	0.0000	0.2533	0.0000
Aspect	−0.0038	0.5141	−0.0132	0.0034	−0.0060	0.1867	−0.1091	0.0000	−0.0590	0.0002	−0.0527	0.0000
Bio1	0.1525	0.0000	0.1530	0.0000	0.0732	0.0039	0.0600	0.1600	0.7653	0.0000	0.1910	0.0000
Bio8	−0.0916	0.0000	−0.0533	0.0000	−0.0338	0.0000	−0.0714	0.0000	−0.0179	0.3616	−0.0619	0.0000
Bio12	0.1581	0.0000	0.0644	0.0000	−0.0074	0.2925	0.2371	0.0000	0.2579	0.0000	0.1364	0.0000
Distance from coast	−0.1955	0.0000	−0.0968	0.0000	−0.0815	0.0000	−0.2991	0.0000	−0.1922	0.0000	−0.1552	0.0000
Slope	0.1481	0.0000	0.0186	0.0064	0.0292	0.0000	0.1684	0.0000	0.0108	0.6301	0.0367	0.0000
adjR^2 best model	0.6192		0.4049		0.2027		0.5248		0.090		0.5484	
AIC best model	5888.7		2848.2		2988.9		6116.5		11,390		2913.4	

**Table 2 plants-12-02640-t002:** The percentage of square grids included in the top 1%, 2.5%, 5%, and 10% hotspots, within each IUCN category, and for all the endemic species.

	Top 1% Squares	Top 2.5% Squares	Top 5% Squares	Top 10% Squares
LC	31.1	51.0	69.0	83.3
NT	41.2	62.9	79.2	90.7
VU	43.4	68.6	82.0	85.6
EN	66.4	82.4	91.7	93.6
CR	100.0	100.0	100.0	100.0
DD	93.3	93.3	100.0	100.0
All categories	46.9	65.3	79.2	88.1

## Data Availability

The data presented in this study are available within the article or in the Appendix A.

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
