# Peer review of "Are Mediterranean Island Mountains Hotspots of Taxonomic and Phylogenetic Biodiversity? The Case of the Endemic Flora of the Balearic Islands"

_plants, 2023, doi:10.3390/plants12142640_

Round 1
Reviewer 1 Report
The paper deals with a very interesting issue for Plants studying the contribution of island mountains to taxonomic and phylogenetic diversity and focusing on endemic flora of Balearic islands.
However, I have some clarification needs.
In the spatila patterns of species richness (line 266 and later) you write about the higher richness of some islands respect to others; have you considered the species-area relation? Explain better please. In Fig. 3 is UICN a mistake? Do you think that elevation is negatively related to anthropic disturbance? Please add some sentences in the discussion. For the introduction and discussion pleaase consider the following papers:
Angiolini, C., Nucci, A., Landi, M., & Bacchetta, G. (2013). Distribution of endemic and alien plants along Mediterranean rivers: A useful tool to identify areas in need of protection?. Comptes Rendus Biologies, 336(8), 416-423.
Di Musciano, M., Zannini, P., Ferrara, C., Spina, L., Nascimbene, J., Vetaas, O. R., ... & Chiarucci, A. (2021). Investigating elevational gradients of species richness in a Mediterranean plant hotspot using a published flora. Frontiers of Biogeography, 13(3).
Author Response
Thank you very much for your comments. As you suggested, we have added a text about the endemic species richness – island area. The relationship is positive (adj. R2=0.92, p-value = 0.0013) but with Menorca below the expected value and Mallorca and Sa Dragonera with slightly higher values.
In Figure 3, UICN was changed to IUCN, thank you for the error detection.
The relationship of the elevation with anthropic disturbance is not an easy issue in the Balearic Islands, due to several historical factors. We agree that is a good point to discuss and we added a paragraph in “Caveats and limitations of the study” in the Discussion section, explaining the difficulty of the relationship between elevation and anthropic impacts:
“We did not include anthropogenic factors as an explanatory variable because, although they are more or less important in some Mediterranean islands [68], the available data at the spatial scale of our study, only includes some modern anthropogenic impacts, but the Balearic Islands has some particularities that difficult the measure, quantification and delineate the spatial extent of these anthropogenic impacts. For example, the human settlements started within the period 2350–2150 BC causing the extinction of several species, the most famous the endemic cave goat Myotragus balearicus Bate, 1909 [106]. The islands had been used by pirates and corsairs and colonized several times by other cultures, and as extreme event, the Cabrera Island was used in its entirety as an open prison in 1809 after the surrender of General Dupont in Baylen, and about 9000 French soldiers were sent to the island of Cabrera transforming completely their landscape [107]. Although the highest elevation areas of the Balearic Islands have a very high concentration of endemic species, their conservation status is very uneven and depends on several intrinsic and extrinsic factors. Initially, it could be assessed that the anthropic disturbance in the highest mountains would be lower compared to lower altitude areas, but this cannot be generalized, and even in some cases there is even evidence to the contrary. The summit of Puig Major (the highest peak in the Balearic Islands) was blown up in 1958 to enable the installation of military radar facilities and an access road was built. Although the Spanish Ministry of Defence has reduced the size of the radar installations and it is involved in the conservation of the natural heritage of the mountain [108] the embankments of the road and the blowing up of the summit buried large areas, causing severe impacts on populations of endemic and rare species [7,47,109]. On the other hand, in recent years the presence of hikers and climbers in the highest areas of Mallorca has increased, which causes the direct (voluntary or involuntary) destruction of specimens of threatened endemic species as well as a loss of quality of the habitat. On the other hand, these higher altitude areas have strong populations of introduced feral goats, which cause severe impacts on the flora and vegetation of the mountain areas of Mallorca [110,111].”
Thank you for the recommendations of these two papers. We have incorporated in the Discussion.
Reviewer 2 Report
Dear Authors,
The introduction is well written and provides a good background for the work. The discussion section requires strong revision because it is currently mixed with the results. There are parts of the discussion that should be included in the results. Put them in the right place.
The manuscript is understandable, but its English needs to be checked.
Minor errors:
L. 78: “scale” instead of “sale”
L. 96: The semicolon is not needed. “..vegetative individuals; [7]) in..”
L. 155, 247, 271, 274, 278, 283, 359, 363, 592: Please use the “×” instead of “x”, and please use space between the units 1 × 1 km Check it through the MS.
L. 164, 166: Please use the “×” instead of “x”. Check it through the MS.
L. 179, 188, 221, 222, 383, 395, 415, 492-493, 500, 537, 554, 583, 585: the format of references is not appropriate to what is expected in the Journal. Please use for example: [number] or Jones et al. [number]. Check it through the MS.
L. 225: a point is missing after the subsp: “..and Magydaris pastinacea subsp femeniesii..”
L. 306-307, 327: the value in brackets should be listed separately from the figure numbering: “..(-0,299; Figure 2; Table 1)..” and “..(one way ANOVA, p=0.8; Figure 3)..”
L. 526: please rephrase the sentence “..squares(Table 2) were are present included in the top 1% squares..”
Fig. 1.: I think it would be better to color code the figure than to explain it in fig. legends (as it illustrated in Fig. 3.). And families may also be worth showing in the figure because not all endemisms can be known to the reader.
Fig. 3.: Figure 3 can be smaller, but the labels and axes should remain resized to a readable size.
Fig. 4.: The size of Figure 4 should be increased, and the axes and labels should be readable and visible. The Island background color should be changed.
Table 2.: The table format is not correct.
Fig. 5.: The background color of the Island must be changed because the other markings are not visible on it.
bekerülnek az eredmények közé. Helyezze Å‘ket a megfelelÅ‘ helyre.
Th
The manuscript is understandable, but its English needs to be checked.
Author Response
Thanks a lot for your revision which helped to improve the structure of the manuscript as well as tables and figures. We have checked and corrected all minor errors you detected, thank you very much:
We have added a legend in the Figure 1 as you suggested (see attached file), however, we discard to include plant families in the Figure 1 because there are 37 families! And representing all families in the figure 1 would complicate the representation and visualisation. Despite this, we agree that plant Family can be crucial for some readers, so we have included a column with plant Family in Table S1.
As you suggested, we rearranged Figure 3 by making it smaller but with larger labels (see attached file).
You and Referee 3 suggested to improve de resolution or size of some or all figures, however, all figures were submitted a high quality (600 dpi) as you can see in the attached file. We will contact to the Editor to comment this issue because we agree that the quality of the images in the draft pdf of our MS is not sufficient. In addition, maps of the Figure 4 were submitted separately so they can be scaled up to a larger size.
In Figure 4 and Figure 5, as you suggested, we changed the Island background colour from dark grey to light grey in order to improve the visibility of all symbols (see attached file). We believe that now everything is more readable.
Table 2 has been formatted in the correct way.
You also commented that “The discussion section requires strong revision because it is currently mixed with the results. There are parts of the discussion that should be included in the results”. We agree that some results were located in the Discussion section, but we believed that some of these results were derived from our primary results and were well placed in the Discussion. Others we believed that did not fit in any of the subsections of our results. However, we have re-read our text bearing in mind your comment and we have rearranged some text from Discussion to Results:
- We put at the end of the “Spatial patterns of phylogenetic diversity” section in the Results, the following sentence related to Figure S3 previously included in the Discussion “Species richness and phylogenetic diversity variables were low correlated except for RE and PD (r=0.72), PE and PD (r=0.69) and WE and CWE (r=0.64) (Figure S3).”
- At the end of the “Conservation priority areas” section in the Results we put the following paragraph related to Table 2 and Table S3 previously included in the Discussion: “The top 1% of grids included almost one square of the 97,4% of all endemic Balearic species (149 species, except Brimeura duvigneaudii (L. Llorens) Rosselló, Mus & Mayol subsp. duvigneaudii, Delphinium pentagynum subsp. formenteranum N. Torres, L. Sáez, Rosselló & C. Blanché, Limonium bianorii (Sennen & Pau) Erben, Orobanche iammonensis Pujadas & Fraga and Polygonum romanum subsp. balearicum Raffaelli & Villar) and with a mean for all species of 46.9% of squares (Table 2) where each species is present in the top 1% squares. In this top 1% squares there are included the 100% of CR species, 66% for EN species and 43% of VU species. With the top 2.5%, 5% and 10% squares all endemic species were included, with a mean for all species distribution of 65.3% (100% for al CR species, 82% for EN species and 69% of VU species), 79.2% (100% for al CR species, 92% for EN species and 82% of VU species) and 88.1% (100% for al CR species, 94% for EN species and 86% of VU species) respectively (Table S5).”
We believe that now there are no results in the Discussion section, or at least there are no primary results that clearly fit Results section.
Thank you.

Reviewer 3 Report
The results of the comprehensive study of the Balearic Islands endemic flora are presented. The authors analyzed the botanical, environmental and phylogenetic data using statistical methods. The obtained results are very important for botany, ecology and conservation. The results supported by figures and tables. It is obvious, that the authors have done a great job during the study and manuscript writing.
I can recommend the paper for the publication after some corrections.
Suggestions to the authors:
Lines 73, 75: Please, explain the abbreviations EDGE and IUCN. Delete the explanation for EDGE on line 217.
Lines 146-148, 181-183, 187 and further: It is necessary to write the authors of the species and genera at the first mention.
Figure 1: Is it possible to improve the resolution of the species name?
Figure 4: It is necessary to enlarge the figure.
Table 1: align in the last two lines.
Table 2: Please, use the Palatino Linotype font.
Figure 5: Please, enlarge the figure.
Line 597: correct Teucrium to Italic.
Author Response
We really appreciate all your comments and we accepted all your suggestions in order to improve the manuscript.
We have explained the abbreviations EDGE and IUCN in the Introduction and deleted the EDGE explanation on line 217.
We have written the authors of the species and genera at the first mention all over the paper, moreover, we have added the authors of all taxa in the table S1. Teucrium in line 597 was changed to Italics.
You and Referee 2 suggested to improve de resolution or size of some or all figures, however, all figures were submitted a high quality (600 dpi) as you can see in the attached file. We will contact to the Editor to comment this issue because we agree that the quality of the images in the draft pdf of our MS is not sufficient. In addition, maps of the Figure 4 were submitted separately so they can be scaled up to a larger size.
Table 1 and Table 2 are checked as you suggested.
Thank you.
